

# Coding potential of circRNAs: new discoveries and challenges

Qingqing Miao[1,2], Bing Ni[3] and Jun Tang[1,2]

[1] Dermatology Department of The First Affiliated Hospital of USTC, Division of Life Sciences and Medicine, University of Science and Technology of China, Hefei, Anhui, China
[2] Department of Dermatology, the 901th Hospital of the Joint Logistics Support Force of PLA Affiliated to Anhui Medical University, Hefei, Anhui, China
[3] Department of Pathophysiology, Third Military Medical University, Chongqing, China

## ABSTRACT

The circular (circ)RNAs are a newly recognized group of noncoding (nc)RNAs. Research to characterize the functional features of circRNAs has uncovered distinctive profiles of conservation, stability, specificity and complexity. However, a new line of evidence has indicated that although circRNAs can function as ncRNAs, such as in the role of miRNA sponges, they are also capable of coding proteins. This discovery is no accident. In the last century, scientist detected the ability of translate in some virus and artificial circRNAs. Artificial circRNA translation products are usually nonfunctional, whereas natural circRNA translation products are completely different. Those new proteins have various functions, which greatly broadens the new ideas and research direction for our research. These series findings also raise questions about whether circRNA is still classified as non-coding RNA. Here, we summarize the evidence concerning translation potential of circRNAs, including synthetic and endogenous circRNA translation ability, and discuss the mechanisms of circRNA translation.

## INTRODUCTION

The classic "central dogma of molecular biology" suggests that the DNA constituent of our chromosomes is transcribed into RNA and subsequently translated into proteins. High-throughput sequencing technology has not only verified the dynamic complexity of gene expression but also revealed the existence of delicate regulatory processes at the RNA level (*Pan et al., 2008*). The RNA form of genetic information serves as the intermediary between DNA and its protein products (*Crick, 1970*); as such, it is believed that levels of RNA are at the core of life's complex functions (*Licatalosi & Darnell, 2010*). At the turn of the century, whole-genome sequencing indicated that while approximately 93% of the DNA in the human genome is transcribed into RNA, only approximately 2% of the DNA sequences encode proteins (*Consortium, 2012*). This finding suggested that there are large amounts of noncoding (nc)RNAs in mammalian cells.

Although the newly discovered ncRNAs were at first largely dismissed as "transcriptional noise", focused investigations began to reveal functional roles in cell biology and many disease types. Researchers' attention has now turned towards defining the roles of ncRNAs

Corresponding authors
Bing Ni, nibing@tmmu.edu.cn, nibingxi@126.com
Jun Tang, jiangzhuyan@stu.ahmu.edu.cn

in regulating and modulating host gene expression (*Meller, Joshi & Deshpande, 2015*; *Peschansky & Wahlestedt, 2014*). The current collective data have allowed the two major groups of ncRNAs—the long (l)ncRNAs and small RNAs, grossly stratified according to size—to be further categorized according to function; these functional subcategories include ribosomal (r)RNA, transfer (t)RNA, small nuclear (sn)RNA, small nucleolar (sno)RNA, PIWI-interacting (pi)RNA, micro (mi)RNA, lncRNA, circular (circ)RNA and transcription initiation (ti)RNA (*Cech & Steitz, 2014*; *Wright & Bruford, 2011*). Among these, the miRNAs and lncRNAs have been extensively studied and confirmed to function in gene transcription through pivotal activities in a versatile regulation network (*Guil & Esteller, 2015*; *Mondal & Kanduri, 2013*).

In 2012, *Salzman et al. (2012)* found the massive presence of circRNAs in eukaryotic cells. Thereafter, the circRNAs have been shown particular stability and functional versatility in vivo. For instances, *Hansen et al. (2013)* firstly reported the natural circRNAs' function as efficient miRNA sponges in both physiological and pathological processes. CircRNAs are products of hnRNA backsplicing and the resulting RNAs represent covalently closed circles, which are devoid of terminal RNA cap structures and poly(A) tails. However, the rapid development of genome-wide translation profiling and ribosome profiling has revealed that a small number of small open reading frames (sORFs) within circRNAs actually have peptide- or protein-coding potential. Therefore, circRNAs have now been demonstrated as capable of translating directly into protein, indicating an intriguing potential to directly function in many processes of life. In this review, we will discuss the most recent progress of the research into the translational capacity of circRNAs and towards defining the underlying mechanisms.

## CIRCRNA BIOLOGY

CircRNAs are single-stranded covalently closed circular RNA molecules generated from a broad array of genomic regions, ranging from intergenic, intronic and coding sequences to 5′- or 3′-untranslational sequences (*Chen & Yang, 2015*; *Memczak et al., 2013*). Two models of circRNA biosynthesis have been proposed, both involving back-splicing catalyzed by the spliceosomal machinery. The first of the two, the "exon skipping" model, begins with classical splicing to generate linear RNA. The downstream exon links to the upstream exon, with one or more exons being skipped; the skipped exons then further back-splice to form precursor circRNAs, which undergo further processing to become mature circRNAs. (B) The second of the two models, the "direct back-splicing" circularization model, is related mostly to complementary motifs; in this, the complementary pairing RNA back-splices to produce a precursor circRNA together with an exon-intron(s)-exon intermediate, and the latter is further processed to produce a linear RNA with skipped exons or which is targeted for degradation (*Ashwal-Fluss et al., 2014*; *Jeck & Sharpless, 2014*; *Lasda & Parker, 2014*) (Fig. 1).

To date, four functions have been defined for the circRNAs. First, circRNAs harbor miRNA complementary sequences, facilitating their combination with and ability to adjust the biological function of a large number of miRNAs by functioning as molecular sponges.

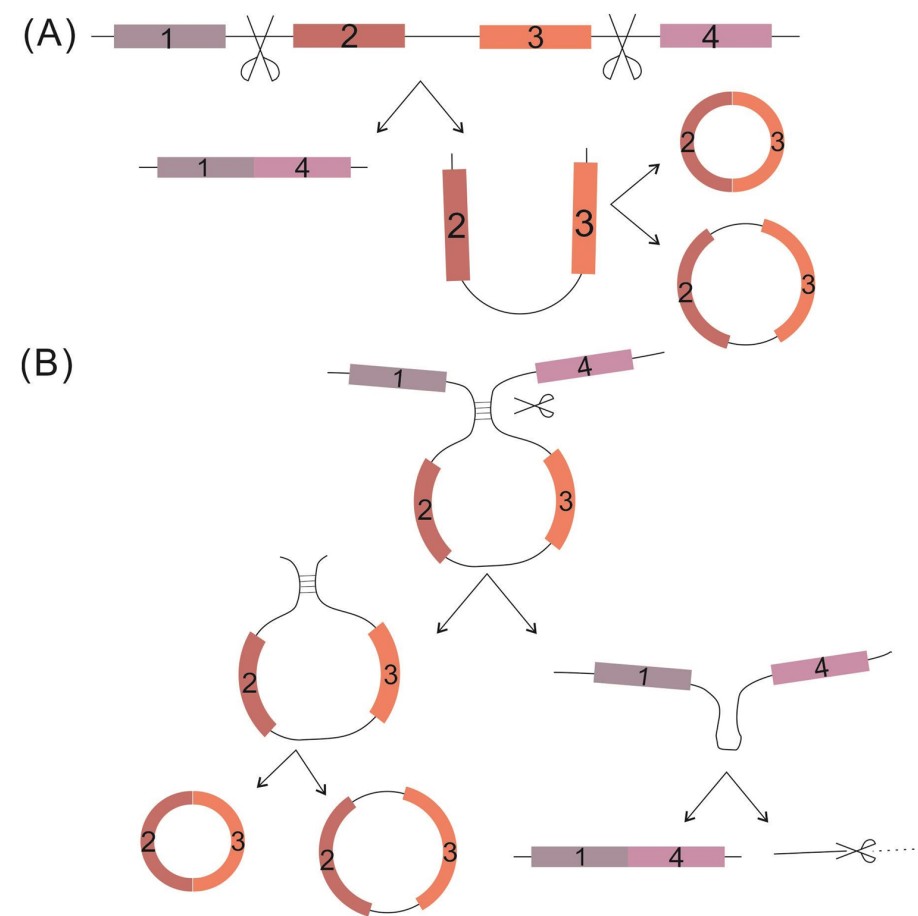

**Figure 1 Proposed circRNA formation models.** (A) The "exon skipping" model. (B) The "direct back-splicing" model. Black thin lines represent intron sequence; colored thick lines represent different exon sequences.

A specific example of this is the circMTO1, which acts as the sponge of miR-9 to suppress hepatocellular carcinoma progression (*Han et al., 2017*). Furthermore, one circRNA may combine with several kinds of miRNAs; for instance, circHIPK3 has been reported to combine with 9 miRNAs (miR-29a, miR-29b, miR-124, miR-152, miR-193a, miR-338, miR-379, miR-584 and miR-654) to synergistically inhibit cell proliferation (*Zheng et al., 2016*). Second, circRNAs can directly regulate transcription, splicing and expression of a parental gene. The exon-intron circRNAs (EIciRNAs) are examples of this regulation, interacting with RNA polymerase II and enhancing transcription of their parental genes (*Li et al., 2015*). Third, circRNAs directly interact with proteins, such as the ternary complex circ-Foxo3-p21-CDK2, which serves to arrest the function of CDK2 and interrupt cell cycle progression (*Du et al., 2016*). However, studies indicate that one circRNA might simultaneously harbor more than one of the above functions, which is evidenced by the finding that circ-Amotl1 can act both as a sponge for miR-17 to promote cell proliferation, migration and wound healing and as a target for protein binding (c-Myc, Akt1 and PDK1)

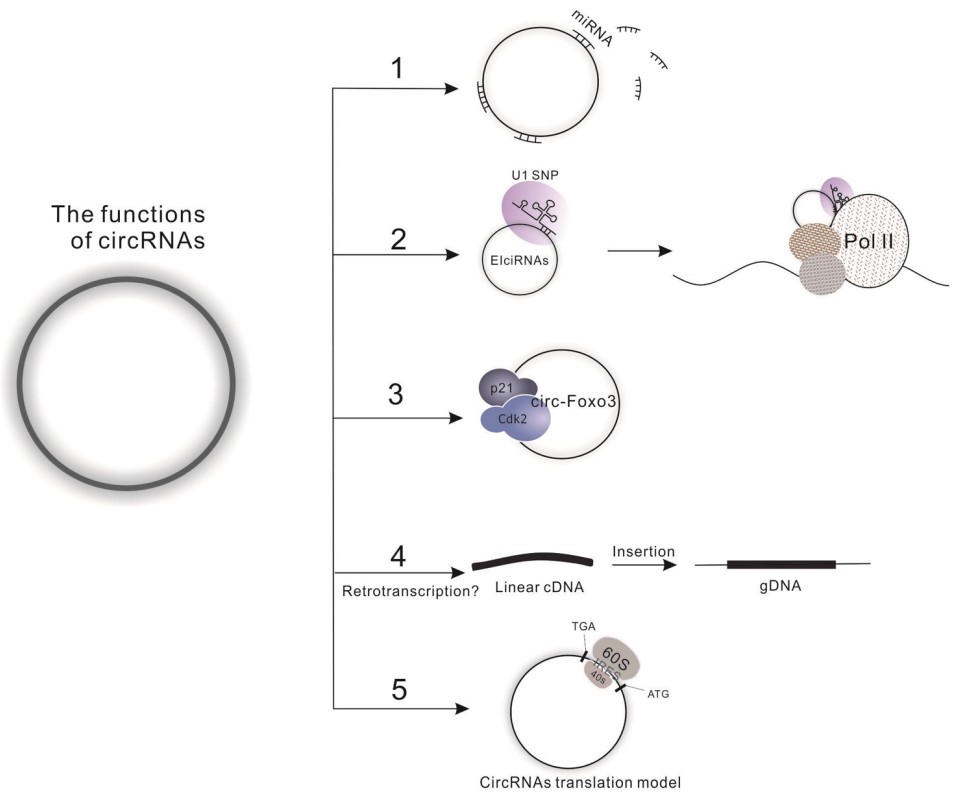

**Figure 2 Functions of circRNAs.** (1) Molecular sponge for miRNA; (2) Regulation of transcription, splicing and expression of parental gene by binding to Pol II; (3) Interaction with proteins; (4) Direct translation of circRNAs.

to promote the proliferation of tumor cells and enhancing cardiac repair (*Yang et al., 2017a*; *Yang et al., 2017c*; *Zeng et al., 2017*). Fourth, Dong et al. developed a computational pipeline (CIRCpseudo), and indicated that stabilized circRNAs could form circRNA pseudogenes by retrotranscribing and integrating into the genome (*Dong et al., 2016*). However, there is only one paper on circRNA's formation of pseudogenes, which does not explain the specific mechanism of it. We need more evidence to prove this idea. More interestingly, the latest research is hinting at a potential fourth function of circRNAs: translation (Fig. 2), which opens a new field for researchers to explore the biological functions of circRNA-derived proteins. For detailed information on the biology of circRNA, please see the review written by Li X et al. (*Barrett & Salzman, 2016*; *Chen, Chen & Chuang, 2015*; *Li, Yang & Chen, 2018*).

## CIRCRNA TRANSLATION POTENTIAL: A CONTROVERSIAL ISSUE EXPLORED UNCEASINGLY

It is commonly believed that mRNAs are the primary controller of cells, carrying out the necessary functions for life. Since the endogenous circRNAs appear to not be associated with polysomes, they presumably lack the potential for translation (*Guo et al., 2014*; *Jeck*

*et al., 2013*). Although this notion has not been definitively disproven, it still attracts scientists' interests in exploring the unknown, hoping to advance the field of research into circRNA translational potency forward from theory to practical knowledge.

## Theoretical basis for direct translation of endogenous circRNAs
### *Molecular structure*

*Internal ribosome entry site (IRES).* It is well known that there are two translation modes, cap-dependent translation and cap-independent translation. The traditional cap-dependent translation accounts for a basal level of protein synthesis under normal growth conditions. In contrast, cap-independent translation contributes to cell proliferation or cellular adaptation/survival when traditional protein synthesis is severely inhibited; this second mode is mediated by the IRES. Therefore, IRES-mediated translation serves as an urgent breakdown maintenance mechanism during cell stress, ensuring basic protein needs are met (*Lang, Kappel & Goodall, 2002*; *Riley, Jordan & Holcik, 2010*); as such, this mechanism is often triggered in conditions of viral invasion, tumor or other human diseases (*Faye & Holcik, 2015*; *Holcik & Sonenberg, 2005*; *Sonenberg & Hinnebusch, 2009*). Thus, it is not surprising that the IRES itself was originally identified by researchers studying the virus parasitic mechanism (*Baird et al., 2006*); since then, comparative sequencing analysis has led to the identification of IRES components throughout the human genome (*Weingarten-Gabbay et al., 2016*). Functional studies have characterized the IRES in mRNA as dependent upon the molecule's special structure, allowing the 40S subunit to avoid assembling directly at the 5′-untranslated sequence (*Sonenberg & Hinnebusch, 2009*).

In 2016, Chen et al. established a circRNA database, circRNADb (http://reprod.njmu.edu.cn/circrnadb) (*Chen et al., 2016*), the first of its kind, summarizing circRNA-encoded protein information based upon 32,914 human exonic circRNAs. Interestingly, their initial explorations of this dataset found ORFs in about half of the circRNAs and IRESs in about half of those; as such, those 7,170 circRNA sequences were considered to fit the characteristic requirements for protein translation capabilities. To date, four types of virus IRES structures are classified with the functional ability to hijack eukaryotic translation machinery, and all work with a common mechanical principle, leading to 80S ribosomal assembly and extension (*Yamamoto, Unbehaun & Spahn, 2017*). However, in eukaryotic mRNAs and circRNAs, the IRES-mediated ribosome assembly mechanism is less well known; only isolated examples of IRESs with known ITAF binding sites or resolved three-dimensional structure are available[35,36].

*RNA modification.* Statistical analyses have estimated that RNA molecules may contain more than 100 distinct modifications (*Gilbert, Bell & Schaening, 2016*). Approximately 16 species of modifications in mRNA have been recognized to date, and the vast majority of these involve the $m^6A$, $\psi$ and $m^5C$ chemical modifications (*Cantara et al., 2011*). The $m^6A$ modification is related to mRNA stability, splicing processing, polypeptide translation and miRNA processing and is correlated with stem cell fate and biological rhythms (*Hoernes, Huttenhofer & Erlacher, 2016*; *Roundtree et al., 2017*; *Squires et al., 2012*). The pseudouridylation modification ($\psi$) serves three main functions, namely, changing the

codon, enhancing the transcript stability and regulating the stress response. To date, only the m$^6$A modification has been verified in circRNAs, wherein it plays a role in promoting translation (*Yang et al., 2017b*). However, research on the m$^5$C modification on ncRNAs has been very limited, though the ncRNA and mRNA have been found to hold thousands of m$^5$C modification sites in recent years (*Hoernes, Huttenhofer & Erlacher, 2016*; *Roundtree et al., 2017*; *Squires et al., 2012*). Therefore, it is speculated that more modification types will be found in both circRNAs and mRNAs with continued research. Such modifications will likely function not only in terms of translation but also in adjusting the functions of circRNAs as ncRNAs. For detailed introduction about circRNAs translation by non-canonical initiation mechanisms, please see the reviews written by *Diallo et al. (2019)* and *Zhang et al. (2020)*.

### Analogous to similar ncRNAs

Recent studies demonstrate that many lncRNAs are able to translate into functional polypeptides. In 2013, *Magny et al. (2013)* found a putative noncoding RNA 003 in 2L (pncr003:2L), including two potentially functional smORFs in the fly's heart, which could translate into bioactive peptides and synergistically regulate cardiac calcium uptake. In 2015, *Anderson et al. (2015)* discovered an annotated lncRNA that translates for a conserved micropeptide- myoregulin (MLN) that functions as a regulator of skeletal muscle physiology. One year afterward, *Nelson et al. (2016)* found that a peptide (named dwarf open reading frame (DWORF)) is encoded by a putative lncRNA. This peptide is mutually exclusive with the other three inhibitors (phospholamban, sarcolipin, and myoregulin) to competitively combine with the SEARC pump to adjust the reuptake of the Ca2$^+$ in muscle. Then, *Matsumoto et al. (2017)* identified a functional novel polypeptide encoded by a lncRNA. This peptide can negatively regulate mTORC1 activation by interacting with the lysosomal v-ATPase in late endosome/lysosome. With deep research, increasingly more lncRNAs with the capacity of translating proteins (peptides) will be explored. As a special type of lncRNAs, we have reason to speculate that the biological significance of coding ability of circRNAs is still to be uncovered.

## Experimental exploration for endogenous circRNA translation in eukaryotic cells

### Early exploration findings

The first indications of a translational role for circRNAs emerged from studies of virus nucleic acids. One of the first observations of a circRNA behaving as a translational template was made with the single-stranded circular RNA genome of the hepatitis δ virus, a satellite virus of the hepatitis B virus; encapsulation of the former by hepatitis B virions was found to result in the production of a single viral protein of 122 amino acids, in a noncanonical manner (*Kos et al., 1986*). In 1995, *Chen & Sarnow (1995)* demonstrated that synthetic circRNAs containing IRES elements were able to correctly translate into polypeptides in rabbit reticulocyte lysate, but those without IRES could not. Furthermore, they speculated that this type of RNA can translate along the RNA circles for multiple consecutive rounds. In 1998, Perriman et al. used plasmids for creating RNA cyclase ribozymes to produce desired circular RNAs that were inserted into the green fluorescent protein (GFP) ORF (finite GFP

encoding) and stop codon-devoid GFP reading frame (infinite GFP encoding) (*Perriman & Ares Jr, 1998*). The authors showed that both circRNAs can directly translate along with GFP in *Escherichia coli* strains and in the meantime, the infinite GFP-encoding RNA could be translated into an extremely long repeating poly-GFP. These findings validated Chen's previous prediction in 1995 (*Chen & Sarnow, 1995*). In 1999, *Li & Lytton (1999)* observed that a circRNA containing NCX1 exon 2 might translate for a protein. It is a pity that they could not detect a protein corresponding exactly to what they predicted from the circular transcript; however, when the circRNAs were made into linear RNAs and transfected into HEK-293 cells, the linear versions of circRNAs were shown to result in the proteins of the expected size of ∼70 kDa, and the transfected cells possessed Na/Ca exchange activity.

Over a decade later, *Wang & Wang (2015)* reported on their construction of an efficient back-splicing circRNA, which could be translated into functional GFP proteins in human and Drosophila cell lines. Furthermore, due to the nuclease resistance characteristics of circRNAs, when the cell was transfected with circRNA, protein production was prolonged for several days. In the same year, *Abe et al. (2015)* provided evidence that circRNAs were translated into infinite FLAG proteins in rabbit reticulocyte lysate and HeLa cells with an infinite ORF in the absence of any particular translation initiation element such as a poly-A tail, internal ribosome entry, or a cap structure. This series of experiments proves that artificial circRNAs with stop codon mutations have a rolling circle amplification (RCA) mechanism to code for long repeating poly proteins. In 2014, *AbouHaidar et al. (2014)* reported a small new virusoid with covalently closed circular (CCC) RNA (220 nt) associated with rice yellow mottle virus that could translate into a 16-kDa highly basic protein. This example is the only one that codes proteins among all known viroids and virusoids. This unique natural supercompact "nano genome" even overlaps its initiation and termination codons to UGAUGA (*AbouHaidar et al., 2014*).

Nevertheless, all these scattered reports, however, are limited to viruses, bacteria, or synthetic circRNA (*Granados-Riveron & Aquino-Jarquin, 2016*) (Table 1), and the translation ability of endogenous circRNAs still requires further exploration.

### Solid evidence for endogenous circRNA direct translation

In 2013, *Jeck et al. (2013)* reported that circRNAs are abundant, conserved and associated with ALU repeats, but there are no detectable levels of exonic circRNAs in the ribosome-bound fraction (via ribosome profiling). One year later, *Dudekula et al. (2016)* raised doubts about this conclusion when they reported their findings from a bioinformatic analysis; IRES regions in circRNAs represented predicted binding sites for RNA binding proteins, including some known to modulate IRES-driven translation.

In 2017, it was finally proved that endogenous circRNAs are capable of directly translating into proteins. By using ribosome footprinting and immunoprecipitation of Drosophila brain tissues, Pamudurti et al. demonstrated that circRNA sequences could be bound by ribosomes including the termination codon (*Pamudurti et al., 2017*). They focused on circ-Mbl from the Mbl gene among all of the ribo-circRNAs and repeatedly verified that circ-Mbl could translate into protein. Through the construction of an overexpression vector, the substitution of the ORF with a split Cherry molecule and target mass spectrometry from the

**Table 1  The published circRNAs with translation potential.**

| CircRNA source | Research model | Translation product | Functions | Reference |
|---|---|---|---|---|
| CircRNAs in viruses and bacteria | Hepatitis δ virus | Protein of 122 amino acids | The hepatitis delta antigen (HDAg) | *Kos et al. (1986)* |
| | A virusoid associated with rice yellow mottle virus | 16-kDa highly basic protein | Unknown | *AbouHaidar et al. (2014)* |
| | *Escherichia coli*: 795-nt circular mRNA | GFP | Fluorescent | *Perriman & Ares Jr (1998)* |
| | HPV16-derived-circE7 | E7 protein | Influencing the development of cancer. | *Zhao et al. (2019)* |
| Artificial circRNAs or synthetic modified RNA | HEK-293 cells | GFP | Fluorescent | *Wang & Wang (2015)* |
| | Rabbit reticulocyte lysate and HeLa cells | FLAG protein (EGF, IGF-1, IGF-2) | Human growth factors | *Abe et al. (2015)* |
| | Rabbit reticulocyte lysate | 23-kDa product | Unknown | *Chen & Sarnow (1995)* |
| | HEK293 cells | GFP, Firefly luciferase, human erythropoietin | Convenient for the author to test | *Wesselhoeft, Kowalski & Anderson (2018)* |
| Endogenous circRNAs | Drosophila: circMbl3 | 37.04-kDa protein | Unknown | *Pamudurti et al. (2017)* |
| | Human: circ-ZNF609 | circ-ZNF609-encoded protein | Unknown | *Legnini et al.(2017)* |
| | Human: circ-FBXW7 | FBXW7-185aa | Cooperates with FBXW7 to control c-Myc stability | *Yang et al. (2018)* |
| | Human: circ-SHPRH | SHPRH-146aa | Guarding against full-length SHPRH protein degradation | *Zhang et al. (2018a)* |
| | Human: CircPINTexon2 | PINT87aa | Inhibiting oncogenes transcriptional elongation | *Zhang et al. (2018b)* |
| | Human: circPPP1R12A | circPPP1R12A-73aa | Promoting the colon cancer pathogenesis and metastasis | *Zheng et al. (2019)* |
| | Human: circβ-catenin | β-catenin-370aa | Stabilizing full-length β-catenin | *Liang et al. (2019)* |
| | Drosophila: circSfl | circlSfl protein | Extending the lifespan of fruit flies | *Weigelt et al. (2020)* |

Drosophila brain circ-Mbl was immunoprecipitated. In the same year, through a screening study of circRNAs related to human, mouse (C2,C12) and a Duchenne muscular dystrophy disease model, Legnini et al. reported that the circ-ZNF609 combined with ribosomes and that its encoded protein was suggested to be involved in the myoblast growth process; however, the circ-ZNF609 was found to be translated at almost two orders of magnitude lower efficiency than that of the linear form (*Legnini et al., 2017*).

Thereafter, *Yang et al. (2017b)* explored circRNA translation ability by the same approach and found that control sequences without IRES were also capable of translating the target protein. These unexpected circRNA translation events were initiated by eIF4G2 and eIF3A and associated with the $m^6A$ modification. When the $m^6A$ modifications were "erased", the target protein translation activity was substantially affected, to the point that it completely disappeared. Ribosome spectrum analysis confirmed that a multitude of
endogenous circRNAs were bound by ribosomes, but whether these circRNAs harbored any IRESs was not examined. Finally, high-throughput sequencing analysis determined that approximately 13% of the total circRNAs carried the $m^6A$ modification. Months later, another independent study showed that circRNAs carry extensive $m^6A$ modifications and are expressed in cell type-specific patterns (*Zhou et al., 2017*). The writing and reading machinery of these $m^6A$ modifications were found to be similar to those of mRNAs (i.e., involving the METTL3/14 and YTH proteins) but were distinctive in their location patterns; the data also suggested that the $m^6A$ modification did not appear to promote degradation of circRNAs as it does for mRNAs. Ultimately, interpretation of these findings indicates that switching the state of $m^6A$ modifications may allow for functional control of circRNAs.

In 2018, *Yang et al. (2018)* reported that the circ-FBXW7 can translate for a new protein FBXW7-185aa during glioma tumorigenesis. Intriguingly, this protein cooperates with FBXW7, which is encoded in their parental genes, to control c-Myc stability and repress cell cycle acceleration and the consequent proliferation. This is the first study to provide definitive evidence of protein translation via circRNA synergy with the protein expression by parental genes and joint function of the proteins. *Zhang et al. (2018a)* further reported that circ-SHPRH, a circRNA containing an IRES-driven ORF, translates into a functional protein. For this process, circ-SHPRH utilizes overlapping genetic codes to create a UGA stop codon, causing translation of the SHPRH-146aa protein. The translated SHPRH-146aa functions as a protector of the full-length SHPRH protein, guarding against degradation by the ubiquitin proteasome and consequently inhibiting cell proliferation and tumorigenicity in human glioblastoma. In the same year, *Zhang et al. (2018b)* found that the 1084 nt CircPINTexon2 which generated by the circularization of exon 2 of LINC-PINT encodes an 87-aa peptide. This peptide (PINT87aa) suppresses glioblastoma cell proliferation in vitro and in vivo by directly interacting with polymerase associated factor complex (PAF1c) and inhibiting the transcriptional elongation of multiple oncogenes.

In 2019, *Zheng et al. (2019)* reported an upregulated circRNA (circPPP1R12A) in colon cancer tissues that could translate a 73-aa protein (circPPP1R12A-73aa). The circPPP1R12A-73aa promotes the proliferation, migration and invasion abilities of colon cancer via activating hippo-YAP signaling pathway. In addition, *Liang et al. (2019)* reported circ β-catenin originated from β-catenin gene locus could promote tumorigenesis. Knockdown of circ β-catenin repressed liver cancer cell growth and migration in vitro and in vivo by inhibiting Wnt/ β-catenin pathway. In terms of mechanism, circ $\beta$-catenin encoded a novel protein (β-catenin-370aa) which shared homologous N-terminus sequence with wild type β-catenin, but it contained a new C-terminus with 9 specific amino acids. The β-catenin-370aa might function as a decoy for GSK3 β, leading to escape from GSK3 $\beta$-induced β-catenin degradation (*Liang et al., 2019*).

Interestingly, *Zhao et al. (2019)* in the same year identified a virus-derived circRNA, HPV16-circE7, which could translate E7 oncoprotein. By constructing various mutant vectors (such as circE7_noATG) for comparison, the authors found that circE7 can also provide the template for E7 oncoprotein translation. Further studies have found that the initiation of circE7 translation may be related to m6A modification and capable of

generating the E7 oncoprotein in a heat-shock regulated manner. Moreover, HPV16 circE7 is essential for the transformed growth of CaSki cervical carcinoma cells and could be regulated by keratinocyte differentiation (*Zhao et al., 2019*).

In 2020, *Weigelt et al. (2020)* showed that circSfl was highly upregulated in all tissues by next-generation sequencing of wild-type and mutant flies. However, circSfl is lack of enrichment of miRNA binding sites in loop, which makes it unlikely acts as a miRNA sponge. Further study verified that circSfl is translated into a small protein that shares the N terminus with full-length Sfl protein. Furthermore, the protein encoded by circSfl and the protein encoded from the linear Sfl transcripts can positively extend the lifespan of female flies (*Weigelt et al., 2020*), indicating the unique role of circSfl in fly life.

Detailed information for the published coding circRNAs is summarized in Table 1.

## CHALLENGES AND PERSPECTIVES

The field of RNA research has continually emphasized the structural and functional versatility of RNA molecules. This versatility has in turn inspired translational and clinical researchers to explore the utility of RNA-based therapeutic agents for a wide variety of medical applications. Several RNA therapeutics with diverse modes of action are currently being evaluated in large late-stage clinical trials, and many more are in the early clinical development stage, including strategies to modulate target gene expression, such as mRNA, siRNA and miRNA (*Sullenger & Nair, 2016*). For instance, mRNA-modified dendritic cells have shown promising and efficient results in clinical trials (*Benteyn et al., 2015*), and siRNA-based therapeutic agents such as bevacizumab (Avastin; off-label use) have shown success for the treatment of wet, age-related macular degeneration in clinical testing (*Garba & Mousa, 2010*). The circRNAs may regulate gene expression through different mechanisms, including direct translation (*Lyu & Huang, 2017*). Therefore, considering their stability and specific expression features, circRNAs with translation potential could represent strong candidates for development as clinical tools to therapeutically manipulate a wide variety of physiologic and pathologic processes. So far, Wesselhoeft et al. have pioneered the transformation of circRNA into robust and stable protein expression in eukaryotic cells. They also considered that circRNA is a promising alternative to linear mRNA (*Wesselhoeft, Kowalski & Anderson, 2018*). But using circRNA as a clinical tool to treat disease remains a challenge, requiring extensive and in-depth research.

In terms of the discovery and exploration of endogenous circRNA translation proteins, we speculate that there will be a large number of circRNAs with translational function gradually discovered. But these need to be supplemented by a large number of experiments, especially the function of those unknown proteins translated by circRNAs. For example, two cases of circRNA translation of small proteins have been found to act as molecular inhibitors or agonist of their mother protein (*Yang et al., 2018*; *Zhang et al., 2018a*). Therefore, studying the function of these small proteins may improve the mechanism of action of some molecules and even serve as a new target for clinical drugs.

The collective evidence to date implies that the translation of endogenous circular RNA into proteins or peptides may be a widespread phenomenon, though the coding potential

of circRNAs previously had been largely disregarded. Therefore, further studies on the translational capacity of circRNAs should be encouraged and should focus on the existing problems, such as the functions and detailed mechanisms of circRNA modifications, the 5′ cap-independent translation of circRNAs and circRNA-derived protein or peptides. The resulting insights will also be helpful towards furthering our understanding of ncRNA functions in general.

### Funding

This work was supported by grants from the National Key Research and Development Project (No. 2016YFA0502203), the National Foundation of China (Nos. 81502728 and 81670534), and the Anhui Provincial Natural Science Foundation (No. 1408085MH149). The funders had no role in study design, data collection and analysis, decision to publish, or preparation of the manuscript.

### Grant Disclosures

The following grant information was disclosed by the authors:
National Key Research and Development Project: 2016YFA0502203.
National Foundation of China: 81502728, 81670534.
Anhui Provincial Natural Science Foundation: 1408085MH149.

### Competing Interests

The authors declare there are no competing interests.

### Author Contributions

- Qingqing Miao performed the experiments, analyzed the data, prepared figures and/or tables, authored or reviewed drafts of the paper, and approved the final draft.
- Bing Ni and Jun Tang conceived and designed the experiments, prepared figures and/or tables, and approved the final draft.

### Data Availability

There is no original, experimental raw data in this literature review.

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
