# Peer review of "Coding potential of circRNAs: new discoveries and challenges"

_PeerJ, doi:10.7717/peerj.10718_

## Round 0.1 · original submission · Minor Revisions

The manuscript has been reviewed by two reviewers who recommend modifications on the text. I would recommend to modify the text to include a more detailed introduction, with a better description of circRNAs and their discovery. Please, pay attention to cite the correct references in the appropriate format.

Reviewer 1 ·

Basic reporting

I read this review with interest. It is focused on subject undergoing intense (and controversial) study, such as the biology of circRNAs and their roles as non-coding but also protein coding RNAs. I found that the the review is of broad and cross-disciplinary interest and within the scope of the journal. Moreover, to my knowledge there are not any similar reviews in this particular subject. However, I found the introduction a little bit scarce, specially for a review paper.

Experimental design

To my view, I found unnecessary the inclusion of this "survey methodology" in a review paper. Moreover, the authors simply describe a series of keywords (such as "noncoding RNAs, circRNAs or protein, among others) and the classical bibliographic databases used to prepare the manuscript. Does it mean that the authors read and analysed ALL the manuscripts retrieved from the searches described in the survey methodology? Because the answer is most likely negative, I suggest a less broad description of the methodology (if totally required by PeerJ). Sources are adequately cited. As mentioned before, I found the introduction a bit scarce, specially regarding the renaissance of circRNAs and the different labs involved in their rediscovery (ie. Salzman Lab, Kjems Lab...). In general, the review is logically organized.

Validity of the findings

no comment

Reviewer 2 ·

Basic reporting

There is an important work regarding the number of literature references they managed. However, several major issues in the bibliography which require rectification were found. In line 77, the authors mentioned circHIPK3. However, it is not referenced as Zheng Q et al; Nature communications 2016. In line 140, the authors mentioned that the IRES were originally identified in virus and to illustrate this they used the reference 40. However, this referenced is not appropriate to illustrate this point and the reference Pelletier J, Sonenberg N. Nature 1988 is more convenient. In line 205, the authors referenced Dawood et al as reference 58. However, this reference is Dudekula BD et al. RNA Biol 2016. In line 215, the authors used Legnini et al. as reference 57 but it is the reference 63. In line 236, the reference is wrong, Zhang et al is reference 64, and it is the properly reference when the authors want to illustrate the SHPRH-146aa as they have done in line 306. Line 247, the authors indicate the reference 65 like Yun et al. However, the reference 65 corresponds to Yang Y et al. The authors mentioned reference 74 (line 286) to illustrate about of the siRNAbased therapeutic agents such as Macugen and bevacizumab (lines 286-287). However, this reference shown that Macugene is a VEGF antagonist aptamer and Bevacizumab is a humanized primary antibody which inhibits VEGF-A. Probably was a typing or editing mistake but the review refers to Bevasiranib as a siRNA-based anti-angiogenic agent proposed for the treatment of wet age-related Macular degeneration (AMD). The reference listed as 63 (line 445) is not on the text. When the authors show the evidence for endogenous circRNA direct translation they did a nice job illustrating with very well referenced examples (circZNF609, circMB1, circSHPRH, and circ FBXW7). However, considering that there are few examples of circRNAs with encode capacity, the examples of circLINC-PINT, which encodes for a PINT87aa peptide controlling of cell proliferation and tumorigenesis (Zhang M et al. Nature Communications, 2018. doi:10.1038/s41467-018-06862-2), and circβ-Catenin which promotes liver cancer cell grown (W.-C. Liang et al. Genome Biology, 2019. doi:10.1186/s13059-019-1685-4) should be properly mentioned and referenced.
The structure of the article conforms to an acceptable format of “standards sections”. Figures and the table are relevant to the content of the article. However, the table 1 should be improved including the circLINC-PINT and the circβ-Catenin RNAs examples. The kind of circRNA in virus, artificial circRNAs or endogenous circRNAs showed in the table 1 should be clearly delimit using a color code or enclose them in boxes. There is a minor type issue, in the column; functions, it is typed FALG reporter instead FLAG reporter.
There is a new review about how the circRNAs are translated by non-canonical initiation mechanisms; L.H. Diallo et al; Biochimie 2019. https://doi.org/10.1016/j.biochi.2019.06.015. However, the computational prediction as a tool to predict future circRNAs with potential to encode peptides was not one of the subjects of this review. Additionally, due to the publication dates so close between the reviews, it is possible that the authors were not aware. Nonetheless, it is strongly recommended to include the reference in the new version of the review

Experimental design

No comment

Validity of the findings

No comment

Additional comments

No comment

---

## Round 0.2 · accepted · Accept

The revised manuscript has been clearly improved.